# Factors Affecting Motivation among Key Populations to Engage with Tuberculosis Screening and Testing Services in Northwest Tanzania: A Mixed-Methods Analysis

**DOI:** 10.3390/ijerph18189654

**Published:** 2021-09-14

**Authors:** Rabia Abeid Khaji, Venance Muzuka Kabwebwe, Annasia Goodluck Mringo, Thomas Faustine Nkwabi, Jacob Bigio, Christina Mergenthaler, Nathaly Aguilera Vasquez, Tripti Pande, Md Toufiq Rahman, Fredrick Haraka

**Affiliations:** 1SHDEPHA+ Kahama, Shinyanga P.O. Box 564, Tanzania; venancez@yahoo.co.uk (V.M.K.); annasiagoodluck@gmail.com (A.G.M.); nkwabi003@gmail.com (T.F.N.); 2McGill International TB Center, Research Institute of the McGill University Health Center, Montreal, QC H3G 1A4, Canada; jacob.bigio@affiliate.mcgill.ca (J.B.); nathaly.aguileravasquez@mail.mcgill.ca (N.A.V.); tripti.pande@mail.mcgill.ca (T.P.); 3KIT Royal Tropical Institute, Mauritskade 63, 1092 AD Amsterdam, The Netherlands; C.Mergenthaler@kit.nl; 4Stop TB Partnership, 1218 Geneva, Switzerland; toufiqr@stoptb.org; 5Ifakara Health Institute, Off Mlabani Passage, Ifakara, Morogoro P.O. Box 53, Tanzania; drharaka@gmail.com; 6Elizabeth Glaser Pediatrics AIDS Foundation, Ursino 395, 2 Mwai Kibaki Road, Morocco, Kinondoni North, Dar es Salaam P.O. Box 1628, Tanzania

**Keywords:** tuberculosis, case finding, engagement, non-engagement, artisanal small-scale miners, female sex workers, key populations, operational research

## Abstract

In northwest Tanzania, many artisanal small-scale miners (ASMs) and female sex workers (FSWs) live in informal communities surrounding mines where tuberculosis (TB) is highly prevalent. An active case finding (ACF) intervention to increase TB case notification was undertaken in two districts. Alongside this, a study was implemented to understand engagement with the intervention through: (1) quantitative questionnaires to 128 ASMs and FSWs, who either engaged or did not engage in the ACF intervention, to assess their views on TB; (2) qualitative interviews with 41 ASMs and FSWs, 36 community health workers (CHWs) and 30 community stakeholders. The mean perceived severity of TB score was higher in the engaged than in the non-engaged group (*p* = 0.01). Thematic analysis showed that health-seeking behaviour was similar across both groups but that individuals in the non-engaged group were more reluctant to give sputum samples, often because they did not understand the purpose. CHWs feared contracting TB on the job, and many noted that mining areas were difficult to access without transportation. Community stakeholders provided various recommendations to increase engagement. This study highlights reasons for engagement with a large-scale ACF intervention targeting key populations and presents insights from implementers and stakeholders on the implementation of the intervention.

## 1. Introduction

Affecting over 10 million individuals every year, tuberculosis (TB) is one of the leading causes of death worldwide [1]. In 2019, there were 2.9 million individuals who were not reported to national TB programs (NTPs), referred to as the “missing millions” [1]. In the same year, the United Republic of Tanzania, hereafter Tanzania, estimated that of the 137,000 individuals who fell ill with TB, 55,800 (41%) were not reported to the NTP [2]. Tanzania also has a high burden of people living with both TB and human immunodeficiency virus (HIV), with an estimated 17% of the global 3.5 million individuals with TB/HIV living within its borders [1].

Shinyanga and Geita are adjoining regions in northwest Tanzania with populations of 1.3 million and 1.7 million, respectively. Each region contains several informal, small-scale mines. These mines employ a large population of artisanal small-scale miners (ASMs) who reside in tents near the mines and often lack access to healthcare services [3,4]. ASMs face occupational hazards that increase their risk of contracting TB, as they are exposed to silica dust, which can cause silicosis [5]. Studies have shown that gold miners with silicosis have almost three times higher risk of contracting TB than miners without silicosis [5].

Surrounding the small-scale mining sites are informal communities served by a provisional workforce including food and alcohol vendors and many female sex workers (FSWs). As the informal communities exist predominantly because of the mines, miners form a substantial proportion of the clients of FSWs operating in these areas. TB prevalence and incidence among FSWs in these settings is under-explored. However, both ASMs and FSWs are at a high risk for human immunodeficiency virus (HIV) infection, which makes them more susceptible to developing TB [4]. In a study of a mining town near Johannesburg, South Africa, more than two-thirds of FSWs were HIV positive [6].

To increase access to services and increase TB case detection among vulnerable populations such as ASMs and FSWs, Service Health and Development for People Living Positively with HIV/AIDS, Kahama Branch (hereafter SHDEPHA+ Kahama) implemented an active case finding (ACF) intervention in the Shinyanga and Geita regions between July 2017 and June 2020 with support from the TB REACH initiative. The ACF intervention targeted ASMs and FSWs by training community health workers (CHWs) and peer-educators (PEs) to educate and mobilise communities for TB detection. This paper presents results from a mixed-methods study conducted alongside the ACF intervention to evaluate the factors affecting motivations for engagement or non-engagement with the intervention by ASMs and FSWs, to gain the perspectives of the CHWs and PEs trained by the intervention and to investigate wider community perspectives on the intervention.

## 2. Materials and Methods

### 2.1. Study Context

Our study was embedded into an ACF intervention targeting ASMs and FSWs. The intervention engaged CHWs and PEs to conduct screening, collect sputum, and refer eligible individuals for HIV testing. The project recruited and trained CHWs and PEs to simultaneously educate the community on TB and provide verbal screening. CHWs and PEs conducted house-to-house screening, worksite screening at mining sites and FSW workplaces, as well as screening at local gathering areas such as bars. Further screening was conducted at educational “moonlight” events organized by the project where attendees were invited to watch infotainment videos on TB and HIV. The presence of one or more TB symptoms (i.e., cough ≥ 2 weeks, fever, unintended weight loss, night sweats and/or hemoptysis) was considered presumptive TB. All consenting individuals with presumptive TB were asked to provide a sputum sample, which was transported to NTP laboratories for bacteriological testing. Individuals who did not consent to providing a sputum sample on-the-spot to the CHWs/PEs were referred to a nearby health facility. For individuals testing positive for TB, treatment was provided free of charge by the NTP. For the purposes of this study, we defined FSWs as women who regularly or occasionally performed sex work.

### 2.2. Study Design

The study utilized a concurrent mixed-methods design with cross-sectional quantitative surveys and qualitative interviews in four parts: (1) quantitative survey of 128 members of key populations who either engaged or declined to engage with the ACF intervention; (2) semi-structured qualitative interviews with 41 of the 128 members of key populations who either engaged or declined to engage with the ACF intervention; (3) in-depth qualitative interviews with 36 CHWs and PEs who worked on the ACF intervention; (4) in-depth qualitative interviews with 30 community stakeholders [Figure 1].

Our study was conducted between December 2019 and June 2020 in informal settlements surrounding small-scale mining sites in Kahama and Msalala districts of Shinyanga region and Bukombe and Mbogwe districts of Geita region, the same communities as for the ACF intervention.

### 2.3. Quantitative Data Collection and Analysis

During the ACF intervention, 7447 FSWs and 22,617 ASMs were screened for the presence of TB symptoms. Our study randomly selected a group of 64 ASMs and FSWs who were screened, tested and diagnosed by the intervention (described hereafter as the engaged group) and a group of 64 ASMs and FSWs who were screened by the intervention but did not agree to give their sputum or attend a hospital or health centre for testing (described hereafter as the non-engaged group). After random selection, ASMs and FSWs from both engaged and non-engaged groups were approached either at health facilities in the four study districts or at their place of work.

Both engaged and non-engaged groups were given a 34-question quantitative survey which aimed to measure the following concepts related to TB disease: perceived susceptibility, perceived severity, perceived benefits, and perceived barriers. These concepts are based on the health belief model, which helps to explain why individuals do or do not participate in health programs [7,8]. Study data collection forms were developed using Kobo Toolbox.

Questions about the four concepts were based on a three-point Likert scale. Answers were given a score (disagree = 1, undecided = 2, agree = 3) based on scoring systems used in previous studies [9,10]. The undecided score was then multiplied by the number of questions relating to each concept to determine the threshold score for each concept (e.g., there were four questions about perceived susceptibility, thus the threshold score for this concept was 8). Scores above the threshold indicated that on average, respondents agreed with the statements in that concept, while scores below the threshold indicated an overall disagreement with statements related to the concept.

Data analysis was performed using Stata 15 SE (StataCorp, College Station, TX, USA). Univariate regression was performed comparing engaged and non-engaged groups for each of the four concepts. Odds ratios (OR) and *p*-values were calculated (*p*-value ≤ 0.05 was considered statistically significant).

### 2.4. Qualitative Data Collection and Analysis

Qualitative data collection took place in three parts. First, a purposive sample of participants from the engaged and non-engaged groups who completed the quantitative survey were selected, with equal numbers selected for each group. Both groups were given a six-question, semi-structured qualitative interview exploring their health-seeking behaviour and reasons for engaging or not engaging with the ACF intervention. Second, a convenience sample of CHWs and PEs, with an equal number of males and females, were given a 19-question in-depth, qualitative interview about their experiences working on the ACF intervention. Finally, to gather information on community engagement with the ACF intervention, a purposive sample of community stakeholders that were essential in the implementation of the intervention (i.e., mine authorities and health facility staff) were selected. Selected community stakeholders were given a separate 19-question in-depth qualitative interview based on four of the nine themes of the Public Health Agency of Canada’s (PHAC’s) Community Capacity Building Tool: participation, knowledge, sustainability and communication [11]. The full sampling process is outlined in Figure 1. Participants were recruited on a voluntary basis.

Questionnaires were administered by members of the study team in Swahili or English in a private location convenient for the participants. Audio recordings from the interviews were transcribed verbatim and translated into English where necessary. Inductive thematic analysis of all qualitative data was conducted using Quirkos v2.4.1 (Quirkos Limited, Edinburgh, UK). A codebook was established for each qualitative data sample by three members of the study team using three interview transcripts. Upon agreeing to the codebook, thematic analysis was conducted on the remaining transcripts in each sample by at least one of the three members.

Qualitative and quantitative data were subsequently triangulated to facilitate understanding and provide a holistic perspective to reasons for engagement/non-engagement with the ACF intervention.

## 3. Results

### 3.1. Quantitative Results

Of the 64 participants in the engaged and non-engaged groups, 37 (58%) and 32 (50%) were female, respectively. The majority of participants were educated to the primary and lower level (50 (79%) in the engaged group and 51 (80%) in the non-engaged group). In both groups, more than half of mine workers had been employed at their current work site for more than 1 year. Other population characteristics are presented in Table 1.

Drillers and stone crushers were significantly more likely to be in the engaged group than FSWs (OR= 2.24; *p* = 0.05 and OR = 4.98; *p* = 0.003, respectively). Participants earning more than 2000 Tanzanian shillings a day were significantly less likely to be in the engaged group than those earning less than 2000 (OR 0.40; *p* = 0.02). There were no other significant differences in characteristics observed between the two groups.

Results of individual Likert scale questions for engaged and non-engaged groups are shown in Table 2. Slightly more respondents in the engaged group agreed that “TB is more serious than most other diseases” and “TB can kill me” than in the non-engaged group (89% vs. 91% and 95% vs. 84%, respectively). Few respondents in either group agreed that “traditional healers can protect me from evil spirits including TB” (17% and 14% in engaged and non-engaged groups, respectively). Approximately one-third of respondents in both groups agreed that “submitting sputum is embarrassing” (34% and 31% in engaged and non-engaged groups, respectively) but more respondents in the engaged group agreed that “getting a TB test result is scary” (39%) than in the non-engaged group (22%). The majority of respondents in both groups believed that “TB medicines can cure TB”, “If I have TB I can get regular checkups” and “It is important to get tested for HIV”.

Overall scores for each concept are shown in Table 3. Perceived susceptibility, perceived severity and perceived benefit scores were above the agreement threshold in both engaged and non-engaged groups. The perceived barriers score was below the threshold in both groups. The mean perceived severity score was significantly higher in the engaged group than in the non-engaged group (*p* = 0.01). None of the other three scores were significantly different between the two groups.

### 3.2. Qualitative Data

Qualitative data was analysed by subgroups. The main themes are presented below.

#### 3.2.1. Factors Influencing Engagement of ASMs and FSWs by the Intervention

In total, 7 (41%) FSWs and 10 (59%) ASMs from the engaged group, as well as 11 (46%) FSWs and 13 (54%) ASMs from the non-engaged group were interviewed. In the engaged group, 8 (47%) respondents were between the ages of 24–30, 4 (24%) were aged 31–40 (4, 24%) and 4 (24%) were aged 41–50. In the non-engaged group, 12 (50%) individuals were between the ages of 31–40, with 6 (25%) aged 24–30. Two major themes emerged from thematic analysis of the qualitative data: health-seeking practices and perception of the intervention.

a.Health-seeking practices

Upon interviewing those that engaged with the intervention, most indicated that a doctor or a pharmacist were the first point of contact when feeling sick, with one individual indicating a preference for traditional medicine. When asked when they would visit a doctor, respondents indicated that they would do so when their symptoms worsened or they were not pleased with pharmacy-bought medication. An individual from the engaged group indicated “normally going to the hospital until I get serious sick” (ASM), while another mentioned: “I do not take medication, I usually go for frequent medical check-up” (ASM). While some individuals in the non-engaged group also indicated visiting hospitals when feeling symptoms of illness, more common answers referred to using traditional medicine such as ginger or seeking out medication from pharmacies: “I usually go buy amoxicillin at the pharmacy store” (ASM). Further, two individuals in the non-engaged group indicated never having visited a hospital. Thus, although some health-seeking behaviour was similar across both groups, there was a tendency for poorer health-seeking behaviour in the non-engaged group. However, quantitative results show that similar proportions of participants in the engaged and non-engaged groups felt they could not get to the health centre because it was too far from the camp site (44% vs. 42%) which could be linked to both groups indicating a preference for health seeking at pharmacies or with traditional healers.

b.Perception of the intervention

While the individuals in both the engaged and non-engaged groups expressed positive interactions with the CHWs/PEs and felt that the TB education was useful, individuals in the non-engaged group indicated a reluctance to provide a sputum sample for various reasons. One notable reason mentioned by some was that they did not understand why they had to provide their sputum: “I didn’t give a sample because I didn’t understand the purpose of asking me to give a sample” (FSW). Other reasons for not providing sputum were: wanted to visit the HIV clinic rather than the TB clinic, collection of samples during times when they were busy or with clients, and having to travel to the clinic to provide sputum. Many individuals gave reasons regarding transportation to testing centers and also offered recommendations indicating “If the TB testing was here [at the mine], the mine people will come for testing” (FSW). This correlates with results presented in Table 2, which show that approximately one-third of respondents in both engaged and non-engaged groups (34% and 31%, respectively) indicated that submitting sputum samples was embarrassing. Further, a higher proportion of respondents in the engaged group (39%) felt that getting a TB test result is scary (22%), although the difference was not significant.

#### 3.2.2. Perspectives of CHWs and PEs

In total, 36 CHWs/PEs were interviewed, of whom 17 (47%) were CHWs, 16 (45%) were PEs and 3 (8%) were unidentified. In total, 21 (58%) were female, 14 (39%) were male and gender was not reported for 1 (3%). Of the respondents, 16 (44%) were between the ages of 24–30 years. Four major themes emerged from the interviews: motivations to occupy CHW/PE positions, role in the intervention and benefits to the community, challenges experienced and recommendations.

a.Motivations to occupy CHW/PE positions

Upon interviewing CHWs/PEs on their experiences with the intervention, the sentiment of serving their community was the most apparent reason for engagement. Many CHWs/PEs felt empowered to continue performing their tasks due to community support. One PE described this as:

“Yes! They [are] always supporting me. They respect my work and [are] always receiving me with enthusiasm, accepting me with good heart, and they always give me information to those people who are symptomatic.” (female, PE)

Further, seeing improved health outcomes of community members was the aspect that most CHWs/PEs liked the most about their job. One individual indicated that they appreciated recognition from the SHDEPHA+Kahama team. The CHWs/PEs reported that the most important benefits the intervention had for them was to increase their knowledge on TB and provide a mechanism to serve their community.

b.Role in the intervention and benefits to the community

The majority of the CHWs/PEs indicated that their main task was to educate individuals on TB and perform screening activities. When asked about methods used to encourage reluctant individuals to seek care, many CHWs/PEs reported that they would continue to educate the individuals on the adverse effects of TB and/or ask their family or family friends to convince them. CHWs/PEs also indicated that engaging with TB survivors to also advocate for the intervention was useful in increasing acceptability of the intervention “by showing them an example of people who have been screened and tested for TB who live in the community” (female, CHW). CHWs and PEs also indicated that the intervention has not only encouraged better understanding of TB for themselves, but for the community as a whole. It has also helped CHWs and PEs to increase access to health services for the community. Further, it was mentioned that there are strong beliefs in the community regarding TB being caused by witchcraft, which interferes with treatment seeking, and also causes testing reluctance. However, CHWs and PEs expressed how the intervention has helped debunk some of these beliefs:
“My community believe in the witchcraft and traditional healer so by giving them education [they] benefit from what I deliver to them.”(male, PE)

In the quantitative survey, 17% of the engaged and 14% of the non-engaged groups indicated believing that “traditional healers can protect me from evil spirits including TB”.
c.Challenges experienced

Almost all CHWs/PEs indicated that they feared contracting TB while on the job. A PE reported this by saying:
“If I will not take precaution, [then] I am at greater risk of been infected with TB from the client [to] whom we provide services.”(male, PE)

Others noted that interacting with drunk and/or abusive individuals was unpleasant and that mining areas were difficult to access. The lack of transportation resulted in CHWs/PEs spending a lot of time reaching the mining areas via bicycle. It was also reported that many individuals refuse to be tested, and that in some situations concerns can be eased by asking village leaders to accompany CHWs as they can encourage community members to undergo testing.
d.Recommendations

CHWs/PEs indicated that providing consistent community awareness will help increase the likelihood of TB testing in mining areas. Further, continuing events beyond project completion would be very beneficial to the community. When asked to comment on their own benefits, CHWs/PEs indicated that a higher monthly allowance and community recognition are most desirable. One CHW also indicated the importance “to continue collaborating with the village chairperson in the provision of TB education to the community” (female, CHW).

#### 3.2.3. Community Stakeholder Perspectives on the Intervention

In total, 30 stakeholders from the community were interviewed. Of those interviewed, 5 (17%) were laboratory staff, 5 (17%) health facility staff, 1 (3%) a clinician, 7 (23%) NTP staff, 4 (13%) administrators of health facilities, 7 (23%) mining site authorities and 1 (3%) was unknown. Three major themes emerged from the interviews: aspects to be improved, strengths and contributions of the interventions and recommendations.

a.Aspects to improve and recommendations

Community stakeholders indicated that the intervention would benefit from providing higher monetary allowances to laboratory and healthcare staff. When asked about aspects that the intervention could improve, community members reported that more supplies such as tents, manpower, electricity and duty allowances would be useful. One community member illustrated this well:
“[There] should be frequent supply of the medical equipment especially the sputum containers, there [was] a time [when] we faced [a] shortage of the sputum containers and by the helping of SHDEPHA [we got] containers from Shinyanga regional.”(female, NTP staff)

Community members also indicated that implementation of GeneXpert in testing sites would decrease wait time to receive test results. When probed on methods to ensure sustainability, all community members mentioned that the continuation of education events such as the moonlight events would be important. Further, they also alluded to continued education for health workers, mining authorities, and CHWs/PEs, including regular refresher trainings to ensure procedures are not forgotten. Some community members also mentioned that further education and more frequent education and educational materials (such as posters in the mining sites and/or radio announcements) are needed for ASMs because they do not always attribute their symptoms to TB, and see them more as consequences of their occupation:
“It [is] hard for the ASM community to know if they have been infected with TB due to the nature of their work, [they] are exposed to dust, even if they show TB symptoms they do not take serious, they think it is because of the dust.”(male, mining site authority)

To encourage acceptance of the intervention, community members suggested involving celebrities to raise awareness and also engaging traditional healers as CHWs.

b.Strengths and benefits to the community

When asked about the strengths of the intervention, community stakeholders highlighted that the CHW/PE training was a major strength. Others noted that staff participation and the willingness of the staff to consistently participate in the intervention was a large strength. Community members attributed sense of ownership as the main reason for success of the intervention as well as the ability to provide health services to the members of the mines. Most respondents indicated that if SHDEPHA+Kahama’s activities ceased, this would be detrimental to the community as TB case notifications would decline, with one community member indicating:
“They managed to have so many indicators that serve the community, much determination in providing services to the community, having the CHW who contribute much to high notification.”(NTP staff, male)

## 4. Discussion

Our mixed-methods study was able to evaluate multiple reasons for engagement and non-engagement in an ACF intervention in a high TB burden setting from the perspective of the community members, CHWs/PEs and those who benefited (or not) from the ACF intervention. This study initially quantitatively explored perceived susceptibility, severity, benefits and barriers of the target population of the intervention (ASMs and FSWs) in regards to TB and TB-related health-seeking. This was followed by a series of qualitative interviews to gain insights into reasons for engagement or non-engagement with the intervention from various community perspectives.

All interviewed groups indicated that they were content with the education provided by the intervention. Community members, ASMs and FSWs indicated that the education services were very helpful in understanding the risks associated with TB. Several qualitative findings indicated that many individuals were reluctant to provide on-the-spot sputum samples and those that did not engage with the intervention highlighted that lack of transportation to visit the health centers deterred them from presenting at health centers to provide sputum samples. This indicates that although collecting on-the-spot sputum samples can be beneficial to increase testing, it is important to consider individuals who may be reluctant to provide samples to CHWs/PEs and prefer facility-based sputum collection.

Approximately one-third of respondents in both engaged and non-engaged groups agreed with the statement that “submitting sputum is embarrassing.” The reasons for this are not clear but may be due to lack of privacy during on-the-spot sputum collection, as many FSWs were approached by CHWs/PEs during busy times at their workplace and may not have wanted their clients to know they were undergoing TB testing. However, as similar proportions of respondents in the two groups agreed with the statement, feelings of embarrassment about submitting sputum did not seem to be an important driver of whether an individual agreed to be tested for TB.

CHWs/PEs also indicated that the community was very receptive to the education provided and would refer individuals with TB-like symptoms to them. However, those that did not engage with the intervention highlighted that lack of transportation to visit the health centre was the main reason for not providing sputum samples. This was further highlighted by community members who indicated that additional supplies such as sputum containers would benefit the sustainability of the intervention. CHWs/PEs also highlighted transportation as a weakness of the intervention, as some individuals were living in very remote, hard to reach areas.

ASMs and FSWs who perceived the severity of the disease to be high were significantly more likely to engage with the intervention. This was also highlighted by community members who indicated that increased sense of ownership of one’s health was a major success of the intervention.

All groups indicated that continuation of education events such as the moonlight events would be beneficial for the community. It is important to note that community members also emphasized the education of mining authorities, health providers and CHWs/PEs. This will ensure that the intervention is not only providing education to the community but regulatory authorities as well. Investing in local events, such as the moonlight events, will enable higher penetration of information and facilitate large-scale education activities. However, duty allowances for the CHWs/PEs should be maintained. While most CHWs/PEs indicated that community recognition and the ability to serve their community was their biggest motivating factor, almost all desired a higher monthly allowance for their services. Future interventions should include such expenses within their budgets and encourage integration of education campaigns into local events to enable larger reach.

Our data demonstrates that engaged and non-engaged groups had similar perceived susceptibility, perceived benefits, and perceived barriers. Although perceived severity had a significantly higher score in the engaged group, the non-engaged group scored above the threshold for agreement. Certain demographic variables, such as being an FSW and having higher income, were inversely associated with engagement. During qualitative interviews, certain participants who did not engage in the intervention indicated lacking time to undergo testing. It could be inferred that those who had higher income work more hours, thus have more difficulties finding time to engage in the intervention. Further, there is some evidence demonstrating that FSWs often experience negative attitudes from healthcare workers and healthcare-related stigma that could in turn cause reluctance to engage in health interventions [12,13].

Another topic mentioned by all groups interviewed was health services from traditional healers in the community. The importance of traditional healers in TB detection has been previously highlighted, and they have been targeted by previous ACF interventions in rural Tanzania [14]. Further, beliefs about witchcraft as the origin of TB have been previously associated with diagnostic delays [15]. Future interventions in this population should integrate traditional healers into screening and diagnostic algorithms to ensure higher coverage and acceptance of the intervention.

One notable strength of this study was its mixed-methods approach, which allowed us to not only collect data on the perceived susceptibility, perceived severity, perceived benefits, and perceived barriers related to TB in key populations in Tanzania, but to also better understand the motivations behind their engagement in this ACF intervention. Further, qualitative interviews allowed us to better understand implementation strengths and weaknesses of the intervention. We were also able to gather holistic perspectives from various stakeholders, which informed us on the specific contributions of the intervention to the community and showcased the importance of maintaining such interventions in mining communities in Tanzania. Other strengths of our study were the ability to engage various different actors of the intervention and to carry out interviews while the intervention was going on, rather than retrospectively.

However, there were a few limitations. Some of the interviewees used the Sukuma language and could not speak Swahili, so translators were required to translate the interviewers’ questions and the interviewees’ responses, with responses then translated again into English. Some language subtleties may therefore have been lost in the two rounds of translation. Further, participants in qualitative interviews were selected in a purposive manner that may have caused some selection bias towards participants who had a more positive experience with the intervention. However, since we included both individuals who engaged with the intervention and those who did not, we believe selection bias was limited. Finally, although respondents in both groups agreed with statements such as “submitting sputum is embarrassing” and “getting a TB test result is scary”, the purposive selection meant that participants with these views may not have been selected for qualitative interviews, and so the reasons for these views may not have been fully explored.

## 5. Conclusions

Our study investigated the reasons for engagement and lack thereof in ACF interventions by ASMs and FSWs. We also documented a wide range of stakeholder perspectives on the implemented intervention and collected valuable information on how to improve future interventions targeting this population. Although our study found some factors associated with decreased engagement, it does not seem likely that perceived susceptibility, perceived severity, perceived benefits, and perceived barriers were the only reasons for not engaging in TB ACF interventions. It will be important for future studies to investigate other reasons that could deter individuals with TB from engaging in ACF interventions in mining communities in Tanzania.

Results from this study may be relevant to other countries in sub-Saharan Africa where miners are at high risk of TB, such as Malawi [16], South Africa [17] and Ghana [18], and to other informal communities around mine sites with large populations of FSWs. In general, compared with the health of ASMs, the health of FSWs in ASM communities remains understudied and should be a focus of future studies. Although care should be taken when extrapolating the results of this study to different settings, interviews with CHWs and community stakeholders about operational characteristics of this large ACF intervention may help to inform the design of other such interventions and may add context-specific detail to a recent study of lessons learned from other TB REACH-funded ACF interventions [19].

## Figures and Tables

**Figure 1 ijerph-18-09654-f001:**
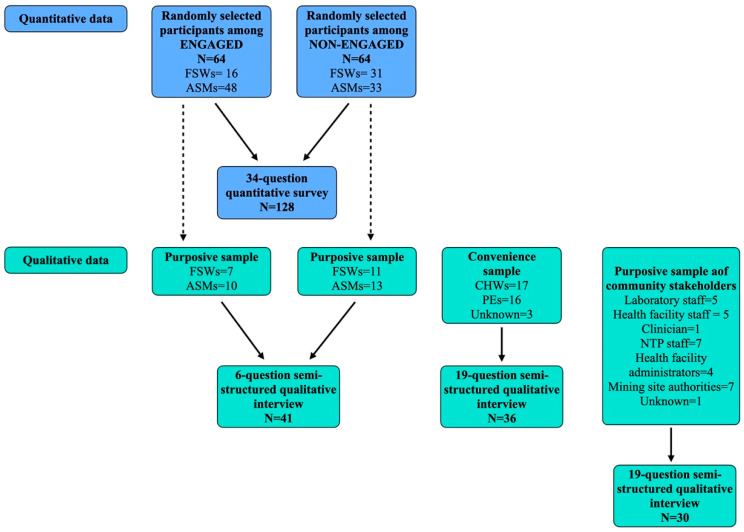
Sampling for quantitative and qualitative data collection.

**Table 1 ijerph-18-09654-t001:** General characteristics of ASM and FSW population (*n* = 128) and univariate analysis for characteristics associated with engagement or non-engagement with the ACF intervention.

	Engaged * (%)	Non-Engaged ** (%)	Odds Ratio	*p*-Value
**Gender**				
Female	37 (58)	32 (50)	1	
Male	27 (42)	32 (50)	1.33	0.42
**Age**				
18–23	6 (9)	7 (11)	1	
24–30	17 (27)	19 (30)	1.04	0.95
31–40	24 (38)	27 (42)	1.03	0.95
41–50	17 (27)	11 (17)	1.80	0.38
**Residence**				
Geita	32 (50)	31 (50)	1	
Shinyanga	32 (50)	31 (50)	1	1
**Level of education**				
Primary and lower level	50 (79)	51 (80)	1	
Secondary school level	13 (21)	11 (17)	1.21	0.68
Higher studies	0 (0)	2 (3)	–	
**Type of work**				
Female sex worker	16 (25)	31 (48)	1	
Drilling	30 (47)	26 (41)	2.24	0.05
Stone crushing	18 (28)	11 (7)	4.98	0.003
**Income level**				
Less than 2000 Tanzanian shillings per day	48 (76)	36 (56)	1	
More than 2000 Tanzanian shillings per day	15 (24)	28 (44)	0.40	0.02
**Length of work in small-scale mining**				
Less than 6 months	2 (3)	3 (5)	1	
6 months–1 year	16 (25)	13 (20)	1.85	0.64
1 year–3 years	18 (29)	17 (27)	1.59	0.53
More than 3 years	27 (43)	31 (48)	1.31	0.78
**Length of work at current site**				
Less than 6 months	6 (10)	7 (11)	1	
6 months–1 year	23 (37)	19 (30)	1.41	0.93
1 year–3 years	17 (27)	21 (33)	0.94	0.59
More than 3 years	17 (27)	17 (27)	1.17	0.81

* Missing data for one participant in level of education, one participant in length of work in small-scale mining, one participant in length of work at current site and one participant in income level. ** Missing data for two participants on residence.

**Table 2 ijerph-18-09654-t002:** Likert scale question responses of engaged and non-engaged groups.

	Engaged			Non-Engaged		
Statement	Agree (%)	Undecided (%)	Disagree (%)	Agree (%)	Undecided (%)	Disagree (%)
Perceived susceptibility questions						
TB is very contagious	48 (75)	6 (9)	10 (16)	41 (64)	12 (19)	11 (17)
Traditional healers can protect me from evil spirits including TB	11 (17)	5 (8)	48 (75)	9 (14)	7 (11)	48 (75)
Someone who is young and healthy can have TB	46 (72)	9 (14)	9 (14)	55 (86)	3 (5)	6 (9)
TB is a danger to me because of my work	49 (77)	9 (14)	6 (9)	54 (84)	3 (5)	7 (11)
Perceived severity questions						
TB is more serious than most other diseases	57 (89)	3 (5)	4 (6)	52 (81)	4 (6)	8 (13)
TB can kill me	61 (95)	3 (5)	0 (0)	54 (84)	4 (6)	6 (9)
If I have TB, I cannot work	53 (83)	0 (0)	11 (17)	39 (61)	5 (8)	20 (31)
Perceived barriers questions						
If you talk to the community health workers, everyone will think you have TB	16 (25)	10 (16)	38 (59)	21 (33)	15 (23)	28 (44)
Submitting sputum is embarrassing	22 (34)	1 (2)	41 (64)	20 (31)	0 (0)	44 (69)
Getting a TB test result is scary	25 (39)	0 (0)	39 (61)	14 (22)	2 (3)	48 (75)
If I have TB I might also have HIV	37 (59)	12 (19)	14 (22)	44 (69)	7 (11)	13 (20)
I cannot get to the health center–it is too far from the camp site	28 (44)	1 (2)	35 (55)	27 (42)	2 (3)	35 (55)
I would not want others to know that I have TB	41 (64)	1 (2)	22 (34)	23 (36)	3 (5)	38 (59)
If I have TB, I will get fired from my job	23 (36)	3 (5)	38 (59)	33 (52)	3 (5)	28 (44)
Perceived benefits questions						
TB medicines can cure TB	59 (92)	5 (8)	0 (0)	55 (86)	3 (5)	6 (9)
TB medication will help someone with TB to be stronger and continue to work	60 (94)	4 (6)	0 (0)	52 (81)	6 (9)	6 (9)
If I have TB I can get regular checkups	61 (95)	0 (0)	3 (5)	56 (88)	3 (5)	5 (8)
It is important to tell others in my community about TB	57 (89)	0 (0)	7 (11)	58 (91)	1 (2)	5 (8)
It is important to tell others to seek medical help if I think they have TB	60 (94)	3 (5)	1 (2)	61 (95)	1 (2)	2 (3)
It is important to get tested for HIV	61 (95)	3 (5)	0 (0)	60 (94)	0 (0)	4 (6)
If I have HIV, I can get medications and live a healthy life	59 (92)	3 (5)	2 (3)	58 (91)	2 (3)	4 (6)

**Table 3 ijerph-18-09654-t003:** Mean scores for Health Belief Model concepts by engaged and non-engaged groups.

Score	Threshold for Agreement	Engaged Mean Score (Sd)	Non-Engaged Mean Score (Sd)	*p*-Value
Perceived susceptibility	8	9.8 (1.2)	9.9 (1.5)	0.45
Perceived severity	6	8.4 (1.2)	7.7 (1.6)	0.01
Perceived barriers	14	13.4 (2.5)	13.2 (3.0)	0.62
Perceived benefits	14	20.3 (1.4)	19.8 (1.94)	0.07

## Data Availability

Not applicable. Data are stored by the NTBLCP and are subject to the organization’s sharing and privacy policies.

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
