# Peer review of "Factors Affecting Motivation among Key Populations to Engage with Tuberculosis Screening and Testing Services in Northwest Tanzania: A Mixed-Methods Analysis"

_ijerph, 2021, doi:10.3390/ijerph18189654_

Round 1
Reviewer 1 Report
I think this is a very interesting article. I feel that it leans a little too into the mines and still underrepresents FSW. Also, were the FSW near the mine camps? A little more information about FSW to balance out the population un-evenness in the article would be great. Finally, I would recommend elaborating a little more in the conclusion. It just ends too abruptly. How can this work be translated into studies in other countries? Maybe pull some global health context into the conclusion to leverage the study's application to other countries - who likely share similar challenges. Still, I think that this article is really interesting and compelling and the mixed methods approach to hold quite a lot of explanatory power by the authors. Excellent work.
Primary recommendation: revise the conclusion to draw in more global health literature and relevancy.
Author Response
Dear reviewer,
We thank you for your helpful comments.
We have edited and added sentences to the introduction to clarify that FSWs work in the areas immediately surrounding the mines and that miners are their main clients, as the informal communities surrounding the mines predominantly exist because of the mines.
A paragraph drawing in more global health literature about ASMs in sub-Saharan Africa, along with a note that the health of FSWs working in ASMs communities is understudied, has been added to the conclusion. Mention has also been made of a study of lessons learned from the implementation of other ACF interventions funded by TB REACH.
Sincerely,
Rabia Abeid Khaji.
Reviewer 2 Report
Recommendation: The manuscript should be publishable with minor revisions.
Comments:
The authors studied the possible factors affecting the motivation of artisanal small-scale miners (ASMs) and female sex workers (FSWs) to engage the tuberculosis screening and testing services. They implemented quantitative questionnaires to ASMs and FSWs and qualitative interviews to ASMs, FSWs, health workers (CHWs), and community stakeholders, then discussed the different perspectives of the intervention and aspects to improve. This study is important for people to understand the major existing hurdles and find a way to improve the acceptance of TB screening and testing service in the mining areas in northwest Tanzania. Overall, this is a well written manuscript with reasonable study design.
However, there are two key points should be addressed:
- The authors mentioned that approximately one-third of respondents both groups agreed that “submitting sputum is embarrassing” and many of them think “getting a TB test result in scary”. They should do more interview and discuss the possible reasons, (i.e. Is their privacy protected if they take the test? Are they scared of losing their job if they got positive results? Financial or social concern?), which would be helpful for figuring out the actual challenges and find a way to motivate people to engage with the test.
- The authors should discuss will the people get affordable treatment if they tested positive for TB, which might be one of the important factors.
Author Response
Dear Reviewer,
We thank you for your helpful comments.
Regarding the statements “submitting sputum is embarrassing” and “getting a TB test result is scary”, we were unfortunately unable to do more interviews as the data collection period ended in 2020. However, a paragraph has been added to the discussion about reasons why submitting sputum might have been embarrassing, particularly for FSWs while they were in their place of work. Additionally, the fact that the qualitative interviews were not necessarily able to fully explore the reasons for participants agreeing with those two statements has been added to the study limitations.
A note has been added in the methods section clarifying that people testing positive for TB receive free treatment from the NTP.
Finally, we thoroughly proof-read the manuscript to improve overall level of English language. Some errors have been corrected throughout the text.
Sincerely,
Rabia Abeid Khaji.